



# A global reference data set for land cover mapping at 10 m resolution

Myroslava Lesiv[1], Steffen Fritz[1], Martina Duerauer[1], Ivelina Georgieva[1], Marcel Buchhorn[2], Luc Bertels[2], Nandika Tsendbazar[3], Ruben Van De Kerchove[2], Daniele Zanaga[2], Dmitry Schepaschenko[1], Linda See[1], Martin Herold[4], Bruno Smets[2], Michael Cherlet[5] and Ian Mccallum[1]

[1]NODES, ASA, International Institute for Applied Systems Analysis (IIASA), Laxenburg, A-2361, Austria
[2]VITO Remote Sensing, Mol, 2400, Belgium
[3]Laboratory of Geo-information Science and Remote Sensing, Wageningen University, Wageningen, 6700 HB, The Netherlands
[4]Remote Sensing and Geoinformatics section, GFZ Helmholtz Centre for Geosciences, Potsdam, 14473, Germany
[5]European Commission-Joint Research Center, Brussels, 1050, Belgium

*Correspondence to*: Myroslava Lesiv (lesiv@iiasa.at)

**Abstract.** This paper presents a unique global reference data set for land cover mapping at a 10 m resolution, aligned with
Sentinel-2 imagery for the year 2015. It contains more than 16.5 million data records at a 10 m resolution (or 165K data records at 100 m) and information on 12 different land cover classes. The data set was collected by a group of experts through visual interpretation of very high resolution imagery (e.g., from Google Maps, Microsoft Bing, ESRI World), along with other sources of information provided in the Geo-Wiki platform (e.g., NDVI time series, Sentinel-2 image time series, geo-tagged photographs, and street view imagery). To ensure high quality and consistency among the experts that collected the data,
regular coordination meetings took place, there were regular quality checks of expert submissions, and comparison with regional land cover maps was undertaken. This extensive reference land cover data set can be used in various applications, e.g., land cover analysis, including mapping and quality verification, ecosystems mapping and modelling, and biodiversity and cropland studies, among others. The data set is available for download at https://zenodo.org/records/14871660.

## 1 Introduction

Land cover mapping is highly dependent on the availability and quality of the training data available (Li et al., 2021). This is especially true when mapping large areas, such as in global land cover mapping, because the high spectral variability of land cover classes in different ecoregions limits the application of training samples to a specific area of interest (Hermosilla et al., 2022). Thus, large, high quality training data sets that have sufficient global coverage and are independent from existing land cover maps are needed. In the past, some data sets have been openly provided, e.g., through the GOFC-GOLD initiative
(Herold et al., 2016), or from crowdsourcing (Fritz et al., 2017), but these have often been collected at much coarser spatial resolutions (e.g., 500 m or 1 km) than are currently needed, without any temporal reference, and are not up to date, having been collected from imagery before the year 2010.



As part of the Copernicus Global Land Service (GCLS-2019), annual global land cover maps at 100 m resolution were developed (CGLS-LC100) (Buchhorn et al., 2020a, b). However, at the start of this process, there were no appropriate open source reference data sets available for training. Thus, there was a critical need to develop a new global reference data set that would fill this gap. More recently, new higher resolution data sets have been published (10-30 m resolution) to support land cover mapping. For example, Stanimirova et al. (2023) published two million training data records collected from Landsat imagery over the period 1984-2020 at a 30 m resolution for use in land cover mapping and change detection, while Brown et al. (2022) collected land cover reference data at a 10 m resolution (to match Sentinel-2) for the development of the 10 m resolution Dynamic World global land cover maps. However, neither of these products existed when the CGLS-LC100 layers were in development.

This paper presents the global reference land cover data set that was developed to support the production of the CGLS-LC100 (Buchhorn et al., 2020a, b), which are comprised of annual global land cover maps and land cover fraction layers that cover the period 2015-2019. The global land cover reference data set is unique as it is independent from existing land cover products, has been collected at a 10 m resolution, and contains more than 16.5 million records at a 10 m resolution (or 165K at 100 m). These records were collected over a 5-year period by a group of experts at the International Institute for Applied Systems Analysis (IIASA). In this paper, we describe the design and collection of the data set, including the thematic attributes, the tools used for data collection, and the quality assurance processes. Together, this reference data set and the two recently released global reference data sets (Brown et al., 2022; Stanimirova et al., 2023) are complementary in their spatial coverage, resolution, and quality, which provides a significant benefit to the land cover mapping community in the production of land cover products.

## 2 Data and methods

### 2.1 Land cover classes and definitions

The global land cover reference data set contains training data labelled with the classes that were used to develop the GCLS-LC100 land cover maps. These classes were defined using the Land Cover Classification System (LCCS) developed by the United Nations (UN) Food and Agriculture Organization (FAO) (Buchhorn et al., 2020a, b). In addition, a few more classes were included to capture uncertainty related to the methodology used to collect the reference data. These classes include burnt areas, fallow land, shifting cultivation, and the class 'Not sure'. Table 1 summarises these classes and their definitions.

Table 1: Class definitions of the land cover reference data set

| No | Class name | Definition |
|---|---|---|
| 1 | Trees | Trees with a height of more than 3 meters. Subpixels were classified as trees when trees fall in the center of a subpixel (10 m x 10 m). |



| 2 | Shrubs | Shrubs are woody perennial plants with persistent woody stems and without any defined main stem being less than 5 m tall. The shrub foliage can be either evergreen or deciduous. Subpixels were classified as shrubs when shrubs fall in the middle of a subpixel. |
|---|---|---|
| 3 | Grassland | Plants without persistent stem or shoots above ground and lack definite firm structure. Tree and shrub cover is less than 10%. These are areas covered by grassland by more than half a subpixel. |
| 4 | Crops | Lands covered with temporary crops followed by harvest and a bare soil period (e.g., single and multiple cropping systems). These crops are harvested at least once per year. Note that perennial woody crops were classified as the appropriate forest or shrub land cover type. |
| 5 | Urban/Built-up areas | Land covered by buildings and other man-made structures that occupy more than a half of a subpixel. |
| 6 | Bare | Lands with exposed soil, sand, or rocks and never have more than 10% vegetated cover during any time of the year. |
| 7 | Burnt | Areas that have been burnt in 2015. It is not possible to assign the land cover type that will be present after the fire. |
| 8 | Water | Permanent fresh or salt-water bodies. |
| 9 | Snow and Ice | Lands under snow or ice cover throughout the year. |
| 10 | Fallow/shifting cultivation | There is not enough information to decide if these were active cropland fields in 2015. It could be fallow land, shifting cultivation, cultivated pastures, etc. |
| 11 | Moss and lichens | Moss and lichen. |
| 12 | Wetlands | Lands with a permanent mixture of water and herbaceous or woody vegetation. The vegetation can be present in either salt, brackish or fresh water. |
| 13 | Not sure | There was not enough information to decide on a land cover type, e.g., no very high-resolution imagery available, no street view imagery, etc. |

## 2.2 Sampling design for the land cover reference data set

The sampling design of the land cover reference data set consisted of two stages:

(1) At the beginning of the data collection process, a global systematic sample was first generated at an interval of approximately 35 km, resulting in around 125K locations.

(2) Once the initial data collection process was completed, additional sample sites were added in areas with low classification accuracy. These sites were identified through visual inspection of intermediate versions of the CGLS-LC100 land cover map that were generated using the initial training data set. In total, 40,000 sample locations were added.



## 2.3 Data collection method

We developed a dedicated branch of the Geo-Wiki (http://geo-wiki.org/) application to collect the land cover reference data at the required resolution of 100 m to match the CGLS-LC100 land cover product. Each 100 m pixel was subdivided into 100 sub-pixels, each with a resolution of 10 m, which is aligned with Sentinel-2 imagery pixels. This allowed for the collection of land cover information at a much finer resolution than the CGLS-LC100 product and for producing the fractional layers. Figure 1 shows a screenshot of the Geo-Wiki interface with the different features and tools highlighted. Each 100 m pixel was laid on top of very high-resolution imagery including Google Maps, Microsoft Bing, and ESRI World imagery for visual interpretation, as well as some MAXAR imagery purchased for use in visual interpretation only. Geo-Wiki also provides access to other information that aids the visual interpretation process. This includes other layers such as regional land cover maps (e.g., CORINE land cover for Europe, land cover maps of Australia, etc.), street view imagery from Mapillary, Normalized Difference Vegetation Index (NDVI) time series derived from Google Earth Engine (GEE) and which can be displayed as graphs, and a time series of Sentinel 2 images that can be retrieved from Sentinel hub. In addition, the location can be displayed in Google Earth Engine for access to historical imagery, geo-tagged photographs as well as Google Street View.

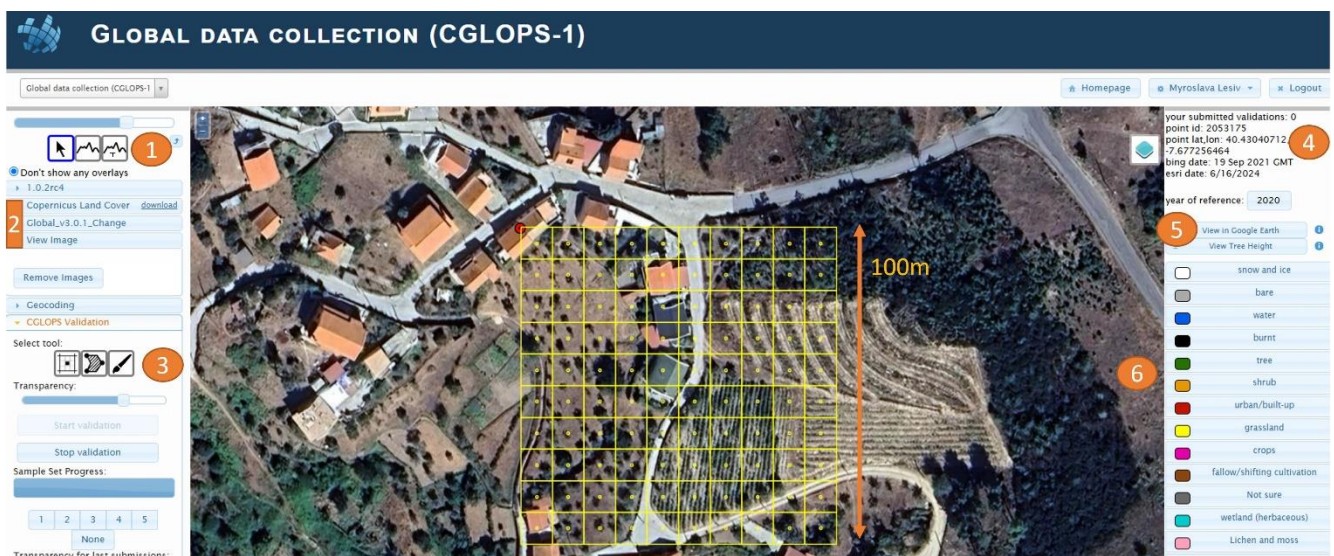

**Figure 1: Screenshot of the Geo-Wiki interface used for collecting the land cover reference data set. The numbered features of the interface are as follows: (1) Additional tools with NDVI time series displayed as graphs and time series of Sentinel-2 images; (2) additional map layers that can be added; (3) drawing tools for annotating individual pixels; (4) general information, such as the number of annotations completed by a user, the coordinates of the current location, and the dates of the Microsoft Bing and ESRI World imagery displayed; (5) a button to generate a kml file that is then displayed in the Google Earth application to allow for access to historical imagery, geo-tagged photographs as well as Google Street View. (6) the land cover legend from which users select when making their annotations. Source of the underlying image: © Google Maps.**

We trained a group of people to interpret each sub-pixel according to the land cover type visible, using the land cover class definitions outlined in Table 1. The training included instructions on how to use the Geo-Wiki tools and to gain a better



understanding of landscapes by looking from above. Over a period of five years, a strong group of 18 land cover experts in visual interpretation was developed. In total, the experts classified 165 696 unique locations (100 pixels at 10 m x 10 m resolution at each location), resulting in 16.5 million data records.

**2.4 Quality assurance processes**

To ensure that the land cover reference data set would be of high quality, the following steps were implemented:


(1) An initial training session was conducted on the use of the Geo-Wiki tools, to explain the different land cover types and to demonstrate how they appear on very high-resolution imagery, e.g., Maxar images at a 50 cm resolution.

(2) We then held regular online meetings to discuss various locations. These meetings took place once per week during the first year and then once every two weeks after that. Their purpose was to reduce any subjectivity related to land
cover interpretation and to better align the interpretations with the definitions provided in Table 1.

(3) In addition to the regular online meetings, we held regular meetings with each individual expert interpreter to do quality checking. This helped assess how well each interpreter understood the task and the land cover definitions. Where necessary, we provided additional training sessions. The quality requirement for an individual interpreter was 90-95%. Thus, out of 100 interpretations that were checked, an interpreter could have made up to 5 to 10
misclassifications, which were mainly random mistakes. If the number of misclassifications was higher, the interpreter was either asked to redo the work or to discontinue further contributions. Therefore, the overall accuracy of the reference data set was 95%.

(4) As an additional quality measure, we compared the expert interpretations with regional land cover maps of high quality (e.g., CORINE land cover, Australian and North American land cover maps, etc.), and we then manually
checked the locations that disagreed.

(5) Finally, locations where visual interpretation was not possible were labelled as 'Not sure' in the data set.

**3 Results and discussion**

**3.1 The reference data set and accuracy assessment**

Figure 2 shows the spatial distribution of the global land cover reference data set, while Table 2 presents the breakdown by
continent and land cover type. The 'Burnt' class is not shown because it makes up a very small number of samples. Some points fell just outside of the continental boundaries or were located in water bodies (e.g., seas and oceans), but all were still used in the development of the CGLS-LC100 land cover product. Figure 3 illustrates the share of land cover reference data points across all land cover types by continent. The predominance of shrubs in Africa and shifting cultivation both in Africa and Asia is clearly visible, while the largest number of points in the 'Not sure' class fell in Asia, possibly because there is less
very high-resolution imagery available for visual interpretation in this region.



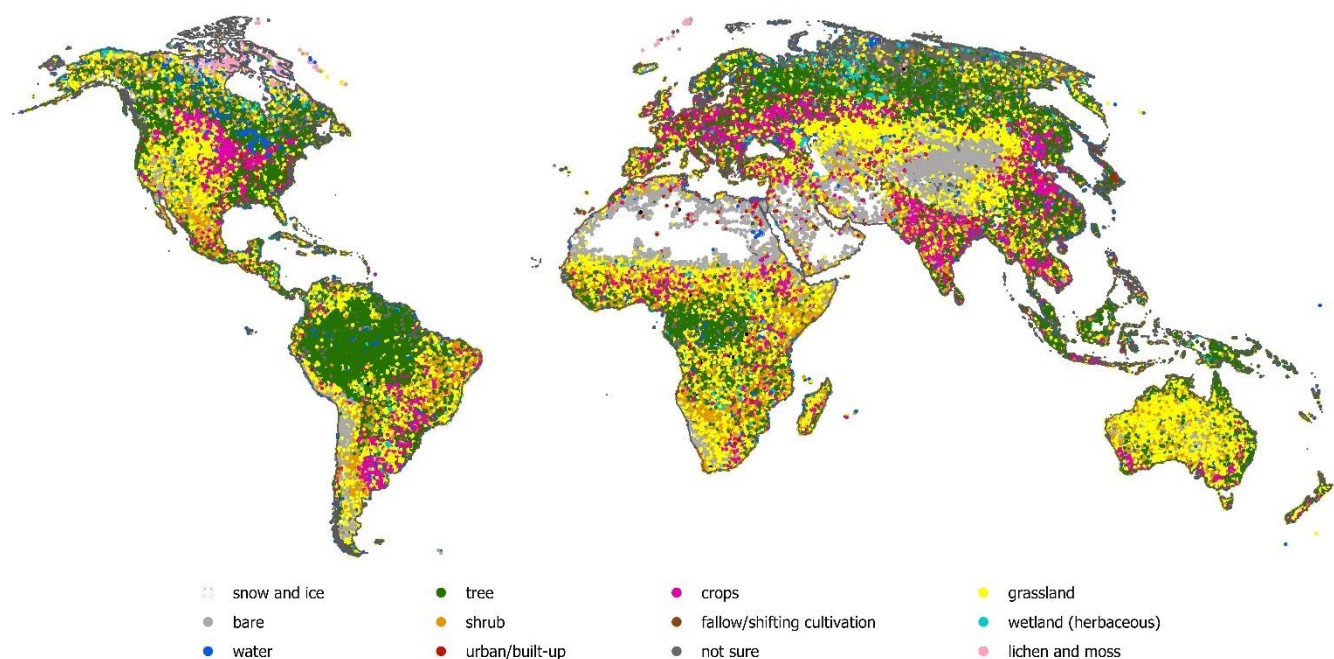

**Figure 2: The spatial distribution of the global land cover reference data set for 2015**

**Table 2: The continental distribution of the number of land cover reference points by land cover class (excluding the 'Burnt' class)**

| Continent | Grassland | Tree | Crops | Shrub | Bare | Fallow shifting cultivation | Urban | Not sure | Burnt | Water | Wetlands | Snow and ice | Lichen and moss | Total |
|---|---|---|---|---|---|---|---|---|---|---|---|---|---|---|
| Africa | 1662618 | 753109 | 486578 | 920984 | 347099 | 34005 | 53319 | 41320 | 10895 | 111927 | 116124 | 7 | 135 | 4538120 |
| Antarctica | 0 | 0 | 0 | 0 | 200 | 0 | 0 | 0 | 0 | 0 | 0 | 100 | 0 | 300 |
| Asia | 1003778 | 1012770 | 476597 | 292744 | 444818 | 35895 | 42695 | 418301 | 674 | 114250 | 167165 | 14625 | 34335 | 4058647 |
| Australia | 439613 | 139628 | 40089 | 122426 | 57331 | 3086 | 1049 | 1046 | 26 | 7423 | 2706 | 0 | 4 | 814427 |
| Europe | 470931 | 577356 | 386216 | 151811 | 38575 | 8038 | 40658 | 46479 | 447 | 65677 | 112262 | 6730 | 4961 | 1910141 |
| North America | 825650 | 877560 | 247338 | 325710 | 149017 | 8385 | 22091 | 110749 | 402 | 180128 | 115057 | 21413 | 38686 | 2922186 |
| Oceania | 21871 | 24555 | 7819 | 2780 | 2348 | 98 | 485 | 562 | 0 | 2312 | 1456 | 207 | 0 | 64493 |
| South America | 641693 | 685191 | 199870 | 274571 | 94509 | 5560 | 8522 | 55335 | 169 | 42407 | 109064 | 1830 | 200 | 2118920 |
| Fell outside | 28820 | 11571 | 3375 | 9368 | 8983 | 74 | 1184 | 3947 | 31 | 54702 | 19752 | 200 | 359 | 142366 |
| Total | 5094974 | 4081740 | 1847882 | 2100394 | 1142880 | 95141 | 170003 | 677739 | 12644 | 578826 | 643586 | 45112 | 78680 | 16569600 |




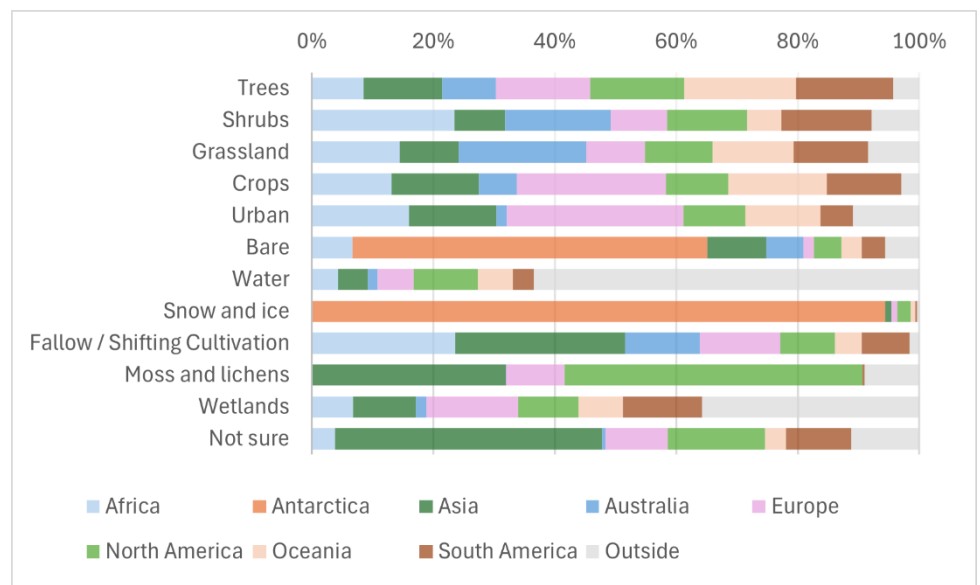

**Figure 3: The share of land cover reference points across land cover classes by continent**

## 3.2 Usage notes

The global land cover reference data set can be used in multiple land cover related applications as follows:

- As a training data set to test various machine learning algorithms to produce land cover maps at various resolutions from 10 m to 100 m;
- As a validation data set for accuracy assessment of land cover maps with resolutions from 10 m to 100 m, with caution recommended in the application since the sampling design is not probabilistic;
- As training and validation data sets for ecosystem mapping and complex modelling of biodiversity; and
- For any other land cover related studies, including land use modelling.

## 3.3 Limitations of the global land cover reference data set

Although the global land cover reference data set proved to be fit for purpose in the development of a 100 m resolution global land cover map (i.e., the dynamic CGLS-LC100m layers), there are a few limitations related to the data set usage at a 10 m

resolution:

(1) The aim was to obtain the correct land cover fractions at a 100 m resolution. For example, if approximately 65% of a 100 m pixel was covered by tree cover, we labelled 65 out of the 100 corresponding 10 m pixels as tree cover. However, there were situations where the trees were located at the intersections of the 10 m pixels. In such cases, tree cover was not the dominant class within individual pixels, yet we still needed to label some of them as "trees" to



match the overall percentage. This introduced some subjectivity to the labelling process regarding which of the 10 m pixels to choose.

     (2) We did not account for potential spatial misalignment between the very high-resolution imagery used for interpretation and the 10 m pixels. This may have resulted in some uncertainty when assigning the dominant class to each 10 m pixel. Potentially, labels of neighbouring 10 m sub-pixels could be considered for this (Xu et al., 2024).

Since not all the samples were checked for consistency, the data set contains up to 5% misclassifications.

## 4 Data availability

The data are openly available from Zenodo (https://zenodo.org/records/14871660, Lesiv (2025)) under a Creative Commons Attribution 4.0 International license.

## 5 Conclusions

The global land cover reference data set at 10 m resolution is a unique collection of high-quality reference data that can support a wide range of land cover applications as well as ecosystem mapping and biodiversity modelling. It contains more than 16.5 million records, each labelled across ~165K locations with one of 12 land cover classes, ranging from tree cover to urban areas. In addition, a 'Not sure' class is included for cases where very high-resolution imagery was not available, there was cloud cover, or there was uncertainty in determining the land cover type. The data set is openly available under a Creative Commons
license.

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
