# Peer review of "A global reference data set for land cover mapping at 10 m resolution"

_Earth System Science Data, 2025_

## Referee Comment (RC1)

**Summary**

This paper introduces a global reference land cover dataset at 10 m resolution based on Sentinel-2 imagery from 2015, containing over 16.5 million data records across 12 land cover classes. The dataset was created through expert visual interpretation of high-resolution imagery (e.g., Google Maps, Bing, ESRI World) along with additional sources from the Geo-Wiki platform such as NDVI time series, Sentinel-2 time series, and geo-tagged photos. The dataset is publicly available via Zenodo and supports applications in land cover analysis, ecosystem modeling, biodiversity, and cropland studies.

**Major comments**

The paper is well written and concise, the methodology is rigorous and sound, the study will contribute to the land cover and land use change community. I see great potential for publication in ESSD. However, there are several shortcomings and clarifications that I strongly suggest the authors address prior to publication. For example, it's unclear from the manuscript how misclassification was determined and how quality of reference data was assessed (see specific comments below).

**Minor comments**

- Table 1 – the term subpixel is not defined. Is a pixel 100 m and subpixel is 10 m? Please clearly define the term in the table. I see the definition in the text on line 73. Note that "subpixels" is spelled inconsistently throughout the text – sometimes it's spelled as sub-pixels and other times as subpixel.
- Line 22 – can you elaborate on how this can be used for biodiversity? It's not obvious to the reader.
- Line 82 – I think the authors mean Google Earth Pro and not Google Earth Engine as Streetview and historical imagery are available on Google Earth Pro.
- Section 2.3 – can you make it explicit that the visual interpretation was done for the year 2015? Could you also elaborate on how you used the land cover maps – I am assuming they were used as ancillary evidence and were not sufficient on their own for labelers to decide? Otherwise, the labels might be reproducing errors in existing land cover datasets. Out of curiosity, was each sample interpreted once?
- Line 93 – how many interpreters were trained? Was this done through a crowdsourcing campaign or were the labelers employees at IIASA/university etc?
- Line 95 – it's unclear if the group of experts is separate from the interpreters trained. Is that a subset of everyone trained? Did experts serve a different function such as reviewing interpreted labels or they were interpreters themselves.

- Line 109 – how was it determined that they were misclassifications? Did a second interpreter check (agree/disagree)?
- Figure 2 – indicates wetlands as herbaceous while Table 1 defines wetlands as either herbaceous or woody. Can you clarify the discrepancy?
- Section 3.2 – could the data be used for a fractional cover classification? Maybe you could list that as a use case as well. Usage in bullet point #2 goes against the good practice for accuracy assessment/validation that has now been widely accepted by the remote sensing community. Maybe instead you could suggest the data be used for a statistical cross validation during the model refinement stages of analysis. https://www.sciencedirect.com/science/article/abs/pii/S0034425714000704
- Lines 146-151 – According to Table 1 "subpixels were classified as trees when trees fall in the center of a subpixel (10 m x 10 m)" and in this portion of the manuscript you are saying "In such cases, tree cover was not the dominant class within individual pixels, yet we still needed to label some of them as "trees" to match the overall percentage." Two things: 1) those two statements are in contradiction, 2) it's unclear what "to match the overall percentage" means, 3) up to this point the impression was that the labeling was done at 10 m resolution and now it appears it was done at 100 m and matched down to 10 m somehow. Can you please clarify? How was 65% cover estimated if not by determining how many of the 100 10 m pixels had tree at the center of the subpixel? This statement at the end seems to contradict the definition in the table.
- Line 155 – still unclear to me how misclassification was determined. See comment above – was it misclassification relative to another interpreter or expert? This is important as you claim that this is a high-quality dataset but it's not exactly clear what metrics were used to determine quality.

**Zenodo comments**
- The difference between validation_id and sampleid is unclear from the description. Seems like they are the same thing.
- Based on Table 1 I thought unique_id would be a value between 1 and 13, however, the values are totally different (e.g., 3027, 3024) and not described anywhere. I think it will be useful to keep these consistent to help users plug this dataset directly into their analyses (which usually require numerical values for classes).

---

## Author Comment (AC2)

Please note that the referees' comments are in black and our responses are in blue.

**Point by point reply below**

**Summary**

This paper introduces a global reference land cover dataset at **10 m resolution** based on **Sentinel-2 imagery from 2015**, containing over **16.5 million data records** across **12 land cover classes**. The dataset was created through expert visual interpretation of high-resolution imagery (e.g., Google Maps, Bing, ESRI World) along with additional sources from the **Geo-Wiki platform** such as NDVI time series, Sentinel-2 time series, and geo-tagged photos. The dataset is publicly available via Zenodo and supports applications in land cover analysis, ecosystem modeling, biodiversity, and cropland studies.

We would like to thank the Referee for their time spent on reviewing the manuscript and for the very useful feedback that we believe has made our manuscript stronger. We provide our point-by-point responses to all the comments below.

**Major comments**

The paper is well written and concise, the methodology is rigorous and sound, the study will contribute to the land cover and land use change community. I see great potential for publication in ESSD. However, there are several shortcomings and clarifications that I strongly suggest the authors address prior to publication. For example, it's unclear from the manuscript how misclassification was determined and how quality of reference data was assessed (see specific comments below).

Thank you for raising the issue of the description of the quality assessment of the reference data set. We have now improved the text; see our point-by-point responses to the comments below. With respect to the major comment on quality, please check our responses on Line 109 and Lines 146-151.

**Minor comments**

Table 1 – the term subpixel is not defined. Is a pixel 100 m and subpixel is 10 m? Please clearly define the term in the table. I see the definition in the text on line 73. Note that "subpixels" is spelled inconsistently throughout the text – sometimes it's spelled as sub-pixels and other times as subpixel.

We have added the definition of the term "subpixel" (which is 10m) to Table 1 and corrected the spelling throughout the text.

Line 22 – can you elaborate on how this can be used for biodiversity? It's not obvious to the reader.

Line 22 refers to the abstract, which has a limited number of words allowed. But we see that we again mention biodiversity in Section 3.2 "Usage notes" (line 140). We have now extended the sentence by adding an example of a possible use such as "an indirect uncertainty assessment of land cover maps used to produce terrestrial habitat types (https://www.nature.com/articles/s41597-020-00599-8)."

Line 82 – I think the authors mean Google Earth Pro and not Google Earth Engine as Streetview and historical imagery are available on Google Earth Pro.

**Thank you for pointing this out. We have now corrected this.**

Section 2.3 – can you make it explicit that the visual interpretation was done for the year 2015? Could you also elaborate on how you used the land cover maps – I am assuming they were used as ancillary evidence and were not sufficient on their own for labelers to decide? Otherwise, the labels might be reproducing errors in existing land cover datasets. Out of curiosity, was each sample interpreted once?

We have made the corresponding changes to Section 2.3 as follows: (1) we have added a sentence to say that the visual interpretation was for the year 2015; and (2) we have added a clarification that the regional land cover maps were used only for evidence.

Line 93 – how many interpreters were trained? Was this done through a crowdsourcing campaign or were the labelers employees at IIASA/university etc?

This was not a crowdsourcing campaign, but rather an internal data collection exercise with interpreters (experts) both from IIASA and some universities. We have added this explanation to the text. In total, we had 18 experts (as stated in the final paragraph of section 2.3).

Line 95 – it's unclear if the group of experts is separate from the interpreters trained. Is that a subset of everyone trained? Did experts serve a different function such as reviewing interpreted labels or they were interpreters themselves.

We can see how this confusion has arisen. Based on our experience, if someone is doing visual interpretation for the first time (and at least for the first year), this person cannot be called an expert. Only after intense training and a substantial amount of time spent on visual interpretations over a year or two, they'd be considered as experts. The interpreters involved in this work have been in the field for more than 5 years so we can now call them experts. To avoid confusion in the manuscript, we refer to people who train and review the visual interpretations as the 'super experts' and all others as the 'experts'.

Line 109 – how was it determined that they were misclassifications? Did a second interpreter check (agree/disagree)?

Super experts at IIASA checked each expert's performance on a weekly basis by reviewing a random subset of 100 locations. Any mistakes that were detected were corrected immediately. If the quality was lower than expected, i.e., less than 90-95%, then either additional individual training took place or the collaboration with this expert was discontinued. We have added this additional explanation to the text.

Figure 2 – indicates wetlands as herbaceous while Table 1 defines wetlands as either herbaceous or woody. Can you clarify the discrepancy?

We have now corrected this figure. It is now consistent and called "wetlands" everywhere. In the C-GLOPS land cover maps, this class is called "herbaceous wetlands" although it may also contain bushes, which could be woody.

Section 3.2 – could the data be used for a fractional cover classification? Maybe you could list that as a use case as well. Usage in bullet point #2 goes against the good practice for accuracy assessment/validation that has now been widely accepted by the remote sensing community. Maybe instead you could suggest the data be used for a statistical cross validation during the model refinement stages of analysis. https://www.sciencedirect.com/science/article/abs/pii/S0034425714000704

Thank you very much for these suggestions. We have added one more usage case on a fractional land cover classification and rephrased the validation case.

Lines 146-151 – According to Table 1 "subpixels were classified as trees when trees fall in the center of a subpixel (10 m x 10 m)" and in this portion of the manuscript you are saying "In such cases, tree cover was not the dominant class within individual pixels, yet we still needed to label some of them as "trees" to match the overall percentage." Two things: 1) those two statements are in contradiction, 2) it's unclear what "to match the overall percentage" means, 3) up to this point the impression was that the labeling was done at 10 m resolution and now it appears it was done at 100 m and matched down to 10 m somehow. Can you please clarify? How was 65% cover estimated if not by determining how many of the 100 10 m pixels had tree at the center of the subpixel? This statement at the end seems to contradict the definition in the table.

**Here is a point-to-point answer to this comment:**

- (1) We have improved the description in Table 1 to make the definitions align. The annotation guidelines were followed as in the first step, the experts labeled the center of each pixel, then they determined if the share of each class was correct at a 100m resolution, and if necessary, small adjustments were made before submitting the final labels.
- (2) The initial purpose of the data set was to train land cover models at a 100m resolution. Therefore, it was important that the fraction of each land cover type inside a cluster (100m x 100m) was corrected since these fractions were derived from the actual number of 10m x10m pixels out of the 100 in total.
- (3) The total number was then determined as the number of pixels out of 100.

We have added a visual example to better reflect this case as shown below. The small adjustment highlighted on the right include two additional pixels annotated as tree cover (central point does not fall on a tree but the majority cover in those 2 pixels is tree cover), and one pixel removed since the tree cover was less than 25%.

**Original location**

Tree cover annotations with small adjustments

Line 155 – still unclear to me how misclassification was determined. See comment above – was it misclassification relative to another interpreter or expert? This is important as you claim that this is a high-quality dataset but it's not exactly clear what metrics were used to determine quality.

The misclassification rate was determined by the experts during the weekly review sessions. Experts at IIASA reviewed 100 random classifications undertaken by each expert on a weekly basis to maintain a high-performance standard. In this way, we are able to detect low-quality performing experts. If the performance quality was low, we had additional individual training sessions, and the experts were asked to redo some of their classifications. If this did not improve the overall quality, we stopped collaborating with these experts. This was an efficient, preventive way to reduce the number of mistakes at the early stages of the data collection process.

**Zenodo comments**

The difference between validation\_id and sampleid is unclear from the description. Seems like they are the same thing.

We have made some changes to the description to clarify this. One sample id can have multiple validation ids because some of the initial submissions were corrected later. In the final data set, we selected only the final correct answers.

Based on Table 1 I thought unique\_id would be a value between 1 and 13, however, the values are totally different (e.g., 3027, 3024) and not described anywhere. I think it will be useful to keep these consistent to help users plug up this dataset directly into their analyses (which usually require numerical values for classes).

| We have made changes to Table 1. We removed the column with ids, which were misleading. |
|-----------------------------------------------------------------------------------------|
|                                                                                         |
|                                                                                         |
|                                                                                         |
|                                                                                         |
|                                                                                         |
|                                                                                         |
|                                                                                         |
|                                                                                         |
|                                                                                         |
|                                                                                         |
|                                                                                         |
|                                                                                         |
|                                                                                         |
|                                                                                         |

---

## Author Comment (AC3)

Please note that the referees' comments are in black and our responses are in blue.

**Point by point reply below**

**General comments:**

This is a short and interesting manuscript describing a reference dataset at 10 meter resolution for land cover and land use mapping. Authors trained experts to judge the land cover type of 10 m resolution sub-pixel through high resolution images, geo-tagged photos and other scientific datasets from multiple sources and compiled results with a global coverage. This product is an important and valuable training/validating data source for fine resolution land cover type or biodiversity mapping. But before I can recommend the paper for publishing on ESSD, I have the following concerns related to how authors present their work.

We would like to thank the Referee for their comments and the feedback on the manuscript. We appreciate the time and effort spent on providing a comprehensive review, which we believe has significantly improved the quality of the manuscript.

**Specific comments:**

1. It is interesting that you provide burnt areas as another land use cover type, but I'm curious about how you can use this information for other studies. Can you provide some examples?

This could be used as a source of verification of fire events that happened around the year 2015.

2. I have a concern about potential time frame mismatch across different products you used. When judging one location, it may be possible that the multiple image or scientific data products you used is obtained from different years, and the land cover type at the exact location might change (e.g., due to urban expansion, deforestation, etc.). I assume this to be one of the uncertainty sources which need experts to put more careful consideration before making decision. How did you handle this?

We checked the locations using the Google Earth Pro application where we could see the exact image dates. If the images were more recent or outdated, we undertook an additional check for changes by visual interpretation of NDVI time series and Sentinel-2

images available from 2015. If there was not enough data available, such locations were labeled as "not sure".

3. I would strongly recommend authors to provide a certain quantitative evaluation on the accuracy of your data product, as the accuracy evaluation to be one of the requirements for publishing your work on ESSD.

We have significantly improved the description of the quality assurance processes (section 2.4). We have added the accuracy number, which falls within the range of 90-95%, and the description of the process of how this number was obtained.

Technical corrections:

Line 64: Where is the start location (lon - lat) of your global systematic sample?

We have added the start location.

Line 67: "in areas with low classification accuracy" Which reference data you used to determine the classification accuracy? Through the assessment information from intermediate versions of the CGLSLC100 land cover map? Please clarify.

We have added further clarification in the text. Areas with low classification accuracy were determined exclusively by visual inspection of the intermediate version of the map. No specific sources of reference data were used to determine the classification accuracies.

Line 68: What is "the initial training data set"? The data produced in step (1)? Need to clarify.

We have now added a clarification to the text to indicate that the initial training data set was produced in step 1.

Line 80: "(NDVI) time series derived from Google Earth Engine (GEE)". Which satellite products did you use to calculate NDVI?

We have now listed the satellite products and added references. This includes Landsat 32-Day Composite (Collection2), MOD13Q1.005 Vegetation Indices 16-Day Global 250m, and PROBA-V C1 L3 Daily at 100m.

Line 81: "a time series of Sentinel 2 images that can be retrieved from Sentinel hub" The same question here. Which product of S2 you used? Or have you created a true color image?

We did not produce any S2 composites by ourselves. We used the ones provided by the Sentinel Hub service, including natural color (bands 4, 3 and 2) and false color images (bands 8, 4 and 3).

Line 95: "18 land cover experts" might be good to acknowledge them if they agree, since they're authors of the dataset.

We have listed all the contributors on Zenodo, and we have now added one more section to the manuscript describing the author's contribution and acknowledging all the experts involved.

Line 109: "Thus, out of 100 interpretations that were checked, an interpreter could have made up to 5 to 10 misclassifications, which were mainly random mistakes." Did you strategically design the distribution of tasks to have, for example, 10% of the dataset assigned to 2 or more interpreters at the same time for accuracy assessment purposes?

The performance of each expert was checked independently and on a regular weekly basis. There was no special design for the distribution of tasks. This was a near real-time continuous process, where at the end of each week, we randomly chose a subset of 100 locations out of all the annotations that were submitted during the past week by each expert, and we reviewed them. We have now improved this paragraph.

Figure 2. I saw desert regions are marked as snow and ice. It's a typo or "missing data" over these regions? If so please use another color to represent regions with "no data". Please clarify and revise the figure.

This is missing data. We have now adjusted the figure and have highlighted snow observations in a different color.

Line 140: "for ecosystem mapping and complex modelling of biodiversity". Can you provide more details about how to map biodiversity?

We have added an example of a possible use such as "an indirect uncertainty assessment of land cover maps used to produce terrestrial habitat types (https://www.nature.com/articles/s41597-020-00599-8)."